# Identification of Entomopathogenic Fungi as Naturally Occurring Enemies of the Invasive Oak Lace Bug, *Corythucha arcuata* (Say) (Hemiptera: Tingidae)

**DOI:** 10.3390/insects11100679

**Published:** 2020-10-07

**Authors:** Marta Kovač, Michał Gorczak, Marta Wrzosek, Cezary Tkaczuk, Milan Pernek

**Affiliations:** 1Croatian Forest Research Institute, Division for Forest Protection and Game Management, 10450 Jastrebarsko, Croatia; martam@sumins.hr; 2Institute of Evolutionary Biology, Faculty of Biology, University of Warsaw, 02-096 Warsaw, Poland; gorczak@biol.uw.edu.pl; 3Botanic Garden, Faculty of Biology, University of Warsaw, 02-096 Warsaw, Poland; martawrzosek@gmail.com; 4Institute of Agriculture and Horticulture, Siedlce University of Natural Sciences and Humanities, 08-110 Siedlce, Poland; cezary.tkaczuk@uph.edu.pl

**Keywords:** *Beauveria pseudobassiana*, *Akanthomyces attenuatus*, *Lecanicillium pissodis*, *Samsoniella alboaurantium*, oak forests, morphology, molecular phylogeny, Croatia

## Abstract

**Simple Summary:**

The oak lace bug (OLB), *Corythucha arcuata*, is a cell sap-sucking insect that is native to North America but has spread rapidly in European countries as an invasive pest. The main hosts are oaks, but it can develop on other forest species as well. Its occurrence is worrying in terms of the cumulative impacts that it could have on forests, as well as the molestation of imago in cities that have been frequently reported. Since the pest is still a new element of a biocoenosis, there is a lack of research on its natural enemies and competitors that could form a potential foundation for biological control strategies. Although there have been reports that lace bugs are quite susceptible to a fungal diseases amid more humid conditions, to date, none of the pathogenic fungi found on OLBs under natural conditions were identified in the literature. In this study, we provide evidence of four entomopathogenic fungi: *Beauveria pseudobassiana*, *Lecanicillium pissodis*, *Akanthomyces attenuatus* and *Samsoniella alboaurantium* that could present a potential as biological control agents against OLBs.

**Abstract:**

The oak lace bug (OLB), *Corythucha arcuata* (Hemiptera: Tingidae), was first identified as an invasive pest in Europe in northern Italy in 2000 and since then it has spread rapidly, attacking large forested areas in European countries. The OLB is a cell sap-sucking insect that is native to North America, with *Quercus* spp. as its main host. Its rapid expansion, successful establishment in invaded countries, and observations of more damage to hosts compared to native areas are most likely due to a lack of natural enemies, pathogens and competitors. In its native area, various natural enemies of OLBs have been identified; however, little is known about the occurrence and impact of OLB pathogens. None of the pathogenic fungi found on OLBs in natural conditions have been identified until now. In this study, we provide evidence of four entomopathogenic fungi that are naturally occurring on invasive OLBs found in infested pedunculate oak forests in eastern Croatia. On the basis of their morphology and multilocus molecular phylogeny, the fungi were identified as *Beauveria pseudobassiana*, *Lecanicillium pissodis*, *Akanthomyces attenuatus* and *Samsoniella alboaurantium*. The sequences generated for this study are available from GenBank under the accession numbers MT004817-MT004820, MT004833-MT004835, MT027501-MT27510, and MT001936-MT0011943. These pathogenic species could facilitate biological control strategies against OLBs.

## 1. Introduction

The oak lace bug, *Corythucha arcuata* (Say, 1832) (Hemiptera: Tingidae), is a cell sap-sucking pest native to North America and a common and important pest of *Quercus* spp. [1,2]. It was recorded in Europe for the first time in 2000 in northern Italy [3], and it has now spread to other countries such as Turkey [4], Switzerland [5], Iran [6], Bulgaria [7], Croatia [8], Hungary [9], Serbia [10], Romania [11,12], Russia [13], Slovenia [14], Bosnia and Herzegovina [15], France [16], Slovakia [17] and Austria [18]. 

This invasive species spread is a result of a combination of different factors, such as translocation with plants or other goods by trade and transport, or movement with humans and traffic vehicles along transport routes or within towns [2]. The main hosts in Europe are oaks, with outbreak populations found most frequently on *Quercus robur* (L.), *Q. petraea* (Matt.) Liebl, *Q. frainetto* (Ten.) and *Q. cerris* (L.) [19]. Besides oaks, *Castanea sativa* (Mill.), *Rosa canina* (L.), *Rubus idaeus* (L.), *R. fruticosus* (L.), *Malus sylvestris* (L.) Mill. and *Ulmus minor* (Mill.) are also mentioned as a host plants [8,20]. 

Both nymphs and adults of OLBs feed directly on leaves by piercing and sucking the nutritious fluid from the cells between the upper and lower epidermis of the leaves, which can cause chlorosis, discoloration and desiccation of the leaf surface, reduction in photosynthesis and even premature leaf fall [7,21,22]. 

It is not yet known if OLBs cause host mortality, but the pest has reached very high population levels in some European countries such as Croatia; this is worrying in terms of the cumulative impacts that it could have on forests [23]. Since its first appearance in eastern Croatia, in lowland stands of pedunculate oak [8] it still occurs with the same intensity and is still spreading to the west and north, causing stress to already susceptible valuable oak stands [14,15,17]. Another aspect of the rising OLB problem in Croatia is accidental contact with imago that fall on human skin. So far, there have been no reports of systemic health effects on humans, but it has been shown that the closely related *C. ciliata* (Aay, 1832) can be an agent of insect-caused dermatosis [24]. Bites and irritation of OLBs have been frequently reported in Croatian cities [25] and mentioned as a new problem in Romania [26].

Their rapid expansion and successful establishment in the invaded countries, and the observation of more damage on hosts in its introduced range than in its original range are probably due to the absence of natural enemies and competitors. The natural enemies of OLBs in their native area include assassin bugs, spiders and lady beetles [27], but there is a lack of research on their impact in European countries, where the pest is still a new element of a biocoenosis. Although there have been reports that lace bugs are quite susceptible to a fungal diseases in humid conditions [28], to date, none of the pathogenic fungi found on OLBs under natural conditions were isolated and identified in the literature.

There are several invasive insect pests in forest ecosystems that have been targets of various management practices, including microbial control with native or introduced entomopathogens [29]. The biological control of pests by using entomopathogenic fungi is an attractive alternative to the use of chemical pesticides, mainly because these fungi are safer for humans, animals, and the environment [30]. More than 700 fungal species from 100 orders are estimated as potential bioagents; however, a majority of important insect pathogens belong to the phylum Ascomycota and order Hypocreales, as well as to Entomophthoromycota, order Entomophthorales [29,31]. The search for virulent fungal isolates against a target host preferably begins with the resident (native) antagonists, because these will be adapted for survival in the target environment. In Croatia, an isolate of naturally occurring entomopathogenic fungus *B. bassiana* that was obtained from *Dendrolimus pini* L. (Lepidoptera, Lasiocampidae) cadavers was tested against *D. pini* healthy larvae and showed promising potential as a biocontrol agent [32].

Biological control has never been seriously attempted with OLBs. However, in vitro experiments which artificially inoculated entomopathogenic fungi spore suspensions on OLB nymphs and adults have been performed, where *B. bassiana* showed the highest mortality rate (80% and 90%) with mycosis values of 77% and 83% [33]. As OLB nymphs and imago live on the underside of leaves, where they actively move, the transmission of pathogens between them shall be theoretically possible. 

The aim of this study was to identify the spectrum of entomopathogenic fungi that naturally occur on OLBs in severely affected oak forests in Croatia. This included both morphological and molecular identification, and phylogenetic analysis, to compare the positions of the obtained strains with the strains obtained from different insects. The species found could form a potential foundation for biological control strategies of OLBs.

## 2. Materials and Methods 

### 2.1. Specimen Collection and Fungi Isolation

*Corythucha arcuata* adults were collected in May 2018, from moss on trees where one part of the population usually overwinters. The chosen area was an infested pedunculate oak forest in the Spačva basin (Croatia), management unit Ceranski lugovi, subcompartment Jelje (45°10′41.3″ N; 18°43′54.9″ E). In the Spačva basin, the fungal infection of *C. arcuata* can be found on up to 18% of individuals per 1m^2^ of moss [34]. To assay the presence and characteristics of entomopathogenic fungi, 15 dead specimens with visible mycosis were brought to the Laboratory of Phytopathology Analysis at the Croatian Forest Research Institute. The naturally infected dead specimens were transferred from the field to the laboratory in Eppendorf tubes and were put in a moist chamber for a further stimulation for fungal sporulation. All cadavers with characteristic symptoms of mycosis were examined under a dissecting microscope (40X) to detect possible contaminants or death caused by other factors. After a few days, fungal isolation was made from developed or sporulating mycelia from *C. arcuata* cadavers on potato dextrose agar (PDA) medium. The cultures were incubated for 1–2 weeks at 25 ± 1 °C, to facilitate growth and sporulation. Then, all isolates were subcultured from a single colony to acquire axenic colonies.

### 2.2. Morphological Characterization

The infected individuals were recognized according to the presence of mycelial growth on the outside of the cadaver [35]. The in vitro cultures were microscopically identified according to the morphology of microstructures [36,37,38]. A characterization of fungal isolates was made by the determination of conidial size and shape, conidiogenous cell and colony morphology. For each isolate, a width and length of 100 conidia and conidiogenous cells from a 14-day-old cultures were measured by using a phase-contrast microscope (Olympus, model BX53, Tokyo, Japan) and magnification 400×, following Inglis et al. [39]. Microscopic images of fungal structures were taken with digital camera (Leica, model MC170 HD, Wetzlar, Germany) mounted on a microscope (Leica, model DM3000 LED, Wetzlar, Germany). Macroscopic images of infected insects were taken under a dissecting microscope (Olympus, model SZX7, Tokyo, Japan) with an Olympus XC30 (Tokyo, Japan) camera and images of fungal colonies were taken with Olympus E-30 (Tokyo, Japan) camera. The twelve fungal cultures were deposited in the Laboratory of Phytopathological Analysis at the Croatian Forest Research Institute (Jastrebarsko, Croatia) and the Institute of Evolutionary Biology at the Faculty of Biology, University of Warsaw under voucher numbers BBNK1, BBNK2, BBC1-BBC10.

### 2.3. DNA Isolation, Polymerase Chain Reaction (PCR) Amplification and Sequencing

Total genomic DNA was isolated from fresh plate cultures of all 12 fungal isolates used in this study by using an ExtractMe Genomic DNA Kit (Blirt S.A., Gdańsk, Poland) following the manufacturer’s instructions. Two molecular markers were amplified for the *Beauveria* strains: the nuclear ribosomal DNA (rDNA) internal transcribed spacer region (ITS1-5.8S-ITS2 = ITS) and the nuclear intergenic region Bloc. For the other Cordycipitaceae strains, seven molecular markers were amplified: internal transcribed spacer (ITS), nuclear rDNA small subunit (SSU or 18S) and large subunit (LSU or 28S), and three nuclear protein-encoding genes, translation elongation factor-1a (TEF), the RNA polymerase II largest subunit (RPB1) and the RNA polymerase II second-largest subunit (RPB2). The primers used are listed in the Table 1. 

PCR conditions for Bloc and TEF were as follow: initial denaturation at 95 °C for 2 min, 34 cycles of 95 °C for 30 s, 56 °C for 30 s, 72 °C for 2 min and final elongation at 72 °C for 10 min. For other loci, the following conditions were set: initial denaturation at 95 °C for 5 min, 34 cycles of 95 °C for 15 s, 52 °C (48 °C for RPB2) for 30 s, 72 °C for 1.5 min and final elongation at 72 °C for 7 min. PCR products were purified with the ExtractMe DNA Clean-up and Gel-out Kit (Blirt S.A., Gdańsk, Poland) and later sequenced with the ABI PRISM Big Dye Terminator Cycle Sequencing Ready Reaction Kit 3.1 (Applied Biosystems, Warrington, UK) with the same primers as in the PCR reaction. The sequences generated for this study are available from GenBank under the accession numbers MT004817-MT004820, MT004833-MT004835, MT027501-MT27510, and MT001936-MT0011943 (Appendix A).

### 2.4. Phylogenetic Analysis

Sequences were assembled using Lasergene v15 (DNASTAR, Inc., Madison, WI, USA), and combined with sequences obtained from Genebank (Appendix A). For the *Beauveria* strains, we used a backbone tree from Imoulan et al. [48], and for the wider Cordycypitaceae phylogeny we used a backbone tree from Zhou et al. [49] with sequences of *Paecilomyces hepiali* Q.T. Chen and R.Q. Dai ex R.Q. Dai et al. (Cordycipitaceae, Hypocreales) added. Multiple sequence alignments were conducted using MAFFT v7.271 [50] and trimmed with trimAI 1.2rev59 [51] or, for protein coding sequences, Gblocks 0.91b [52]. The trimmed alignments were partitioned according to molecular markers, and, in case of protein-coding loci, the codon position was also partitioned. Maximum likelihood trees were generated with RAxML-NG v. 0.8.0 [53].

## 3. Results

### 3.1. Isolation and Morphological Identification of Fungi

A total of 12 fungal isolates were obtained from naturally infected *C. arcuata* adults collected from the field. On the basis of morphological features, we identified these species as *Beauveria pseudobassiana* (S.A. Rehner and Humber), *Lecanicillium pissodis* (Kope and I. Leal), *Akanthomyces attenuatus* (Zare and W. Gams) (Spatafora, Kepler and B. Shrestha) and *Samsoniella alboaurantium* (G. Sm.) (Mongkols., Noisrip., Thanakitp., Spatafora and Luangsa-ard) (Figure 1a–d). *B. pseudobassiana* was the most commonly detected fungus with eight isolates altogether (BBC1, BBC2, BBC3, BBC4, BBC6, BBC8, BBC10 and BBNK1). Isolates BBNK2 and BBC7 were identified as *L. pissodis*, isolate BBC9 as *A. attenuatus*, and isolate BBC5 was identified as *S. alboaurantium*. The morphological characteristics of the fungal species used in this study are summarized in Table 2.

### 3.2. Phylogenetic Analysis

As shown in Figure 2, the strains BBC1 to BBC4, BBC6, BBC8, BBC10 and BBCNK1 clustered with reference *B. pseudobassiana* strains. Strains BBCNK2, BBC7 and BBC9 formed a weakly resolved clade with species of *L. pissodis*, *A. muscarius* and *A. attenuatus*, and their identities were clarified according to their morphologies. Strain BBC5 clustered with *S. alboaurantium* (Figure 3).

## 4. Discussion

In this study, we report the evidence of four entomopathogenic fungi occurring on invasive OLBs found in an infested pedunculate oak forest in eastern Croatia. On the basis of their morphologies and multilocus molecular phylogenies, an identification of naturally occurring entomopathogenic fungi was performed. This is the first report of the occurrence of *B. pseudobassiana*, *L. pissodis*, *A. attenuatus* and *S. alboaurantium* on OLBs in oak forests.

*Beauveria* species are the most common cosmopolitan insect-pathogenic fungi which parasitize over 700 insect species [48,54]. Until now, five species (*B. bassiana*, *B. brongniartii*, *B. caledonica*, *B. varroae* and *B. pseudobassiana*) have been documented in Europe [53]. *Beauveria pseudobassiana* is an entomopathogenic fungus with a worldwide distribution, and it can be pathogenic to a wide range of hosts [37,55,56]. Recent analyses of soil samples in Slovakia have indicated that *B. pseudobassiana* prefers forest habitats over field crops or meadows [57], assumed to be due to its better adaptation to forest ecosystems. It was first described, i.e., distinguished from *B. bassiana*, in 2011 by DNA-based phylogenetic analysis. Since these two species are morphologically similar and could only be differentiated by *B. pseudobassiana* possessing slightly smaller conidia, it was believed that their distinction is only possible through DNA sequence data analysis [37]. More recent research morphologically also separates them by their colony color, where the *B. bassiana* colony was described as being white, and *B. pseudobassiana* as being yellowish-brown [49], which coincides with the findings of this research. Moreover, we noted that our *B. pseudobassiana* isolates slightly changed the color of the nutrient medium to light purple during their growth, which was not recorded in previous studies. This is probably due to some species or strains of *Beauveria* being able to produce specific secondary metabolites that can change the color of some culture media [58,59].

Entomopathogenic fungi species belonging to the genus *Lecanicillium* have a global distribution, occur on a diverse range of insect species, and have potential for development as effective biological control agents against a number of plant diseases, insect pests and plant-parasitic nematodes [60]. For the species *L. pissodis* and *L. attenuatum*, the ITS sequences were identical, but in 2005, they were distinguished by a β-tubulin gene sequence. They can also be discriminated from each other and from other *Lecanicillium* species by differences in morphology [61]. In 2015, some species of *Lecanicillium*, including *L. attenuatum*, fell within the genus *Akanthomyces* [62], and according to Kepler et al. [63], *L. attenuatum* was renamed *Akanthomyces attenuatus*. Despite dubious points in the phylogeny for strains BBNK2, BBC7 and BBC9 in this study, their identities were resolved based on their different morphological characteristics. According to their growth rates in culture, phialides morphologies and conidial shapes and sizes, the strains BBNK2 and BBC7 were identified as being *L. pissodis*, and strain BBC9 was identified as being *A. attenuatus*.

At the initial stages of molecular identification, the ITS sequence of the BBC5 strain was found to be highly similar to that of the *Paecilomyces hepiali*-type strain (99.8% identity). Because of this, we decided to add *P. hepiali* to our phylogenetic analysis, and it clustered as an outmost branch of the *Samsoniella* clade. Although *P. hepiali* is intensively researched as medicinal fungus, there is some uncertainty about its taxonomical placement. Only a few sequences are available from ex-type cultures or wrongly designated neotypes [64,65]. Moreover, *P. hepiali* has not been included in any major molecular phylogenetic studies on Cordycypitaceae. As far as we know, this study is the first to demonstrate that *P. hepiali* and *Samsoniella* are closely related. *Samsoniella* was segregated from *Lecanicillium s.l.* and, based on molecular evidence in 2018, *S. alboaurantium* was originally identified as *Penicillium alboaurantium* by Samson [66], which was affiliated to this newly established genus [67]. 

## 5. Conclusions

Since reports that are related to natural enemies of OLBs are quite limited, especially considering the pathogenic nature of the fungi towards OLBs, the findings of this study provide a good foundation for similar research in the future. Future studies are still required in order to evaluate the potential of these fungi as a group of biological control agents against OLBs. This should include testing their efficacy, both in laboratory conditions and in their natural habitat in order for them to be involved as a component in control strategies.

## Figures and Tables

**Figure 1 insects-11-00679-f001:**
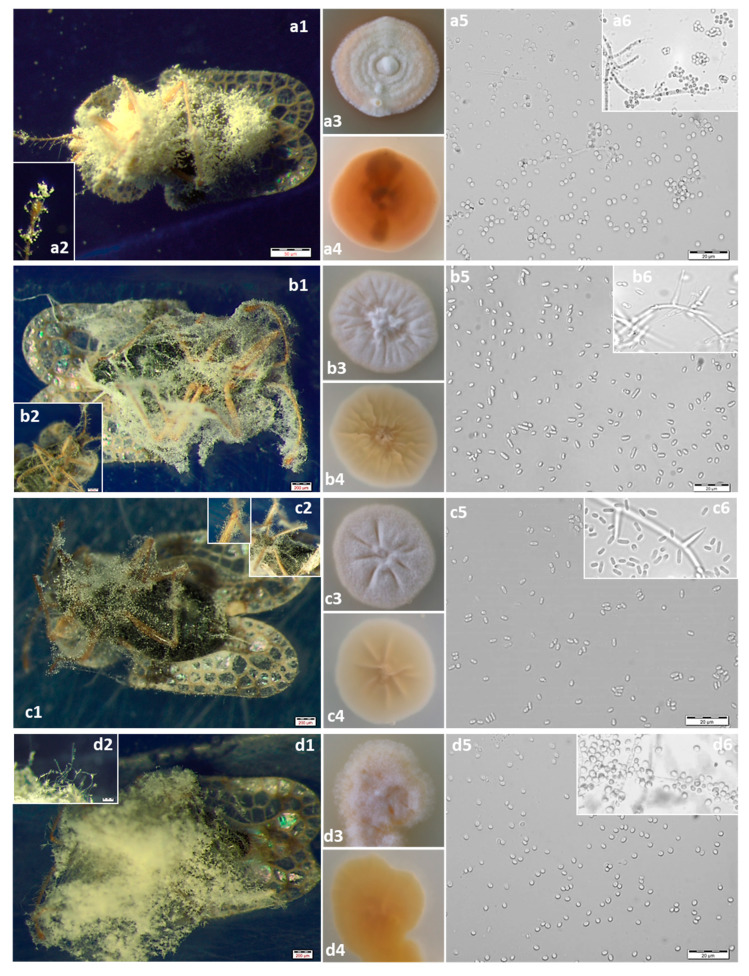
(**a**) *Beauveria pseudobassiana*: (**a1**) Infected adult of *C. arcuata*. (**a2**) Conidiophores. (**a3**) Colony on potato dextrose agar (PDA) at 14 d. obverse. (**a4**) Colony on PDA at 14 d. reverse. (**a5**) Conidia. (**a6**) Conidiogenous cells. (**b**) *Akanthomyces attenuatus*: (**b1**) Infected adult of *C. arcuata*. (**b2**) Conidiophores. (**b3**) Colony on PDA at 14 d. obverse. (**b4**) Colony on PDA at 14 d. reverse. (**b5**) Conidia. (**b6**) Conidiogenous cells. (**c**) *Lecanicillium pissodis*: (**c1**) Infected adult of *C. arcuata*. (**c2**) Conidiophores. (**c3**) Colony on PDA at 14 d. obverse. (**c4**) Colony on PDA at 14 d. reverse. (**c5**) Conidia. (**c6**) Conidiogenous cells. (**d**) *Samsoniella alboaurantium*: (**d1**) Infected adult of *C. arcuata*. (**d2**) Conidiophores. (**d3**) Colony on PDA at 14 d. obverse. (**d4**) Colony on PDA at 14 d. reverse. (**d5**) Conidia. (**d6**) Conidiogenous cells.

**Figure 2 insects-11-00679-f002:**
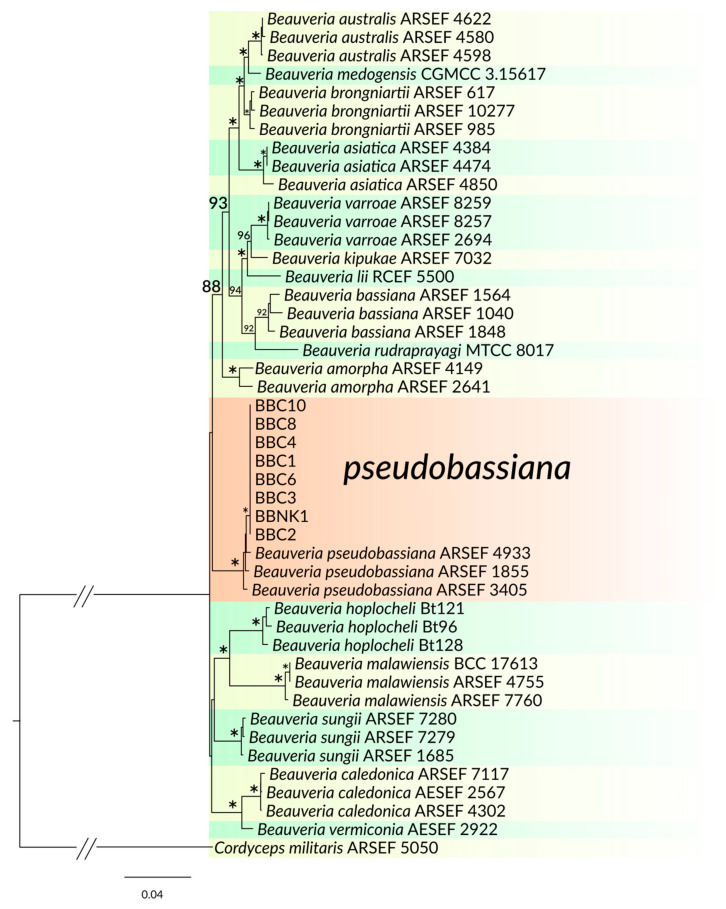
Maximum likelihood phylogenetic tree of selected isolated strains and related *Beauveria* species based on five loci (internal transcribed spacer (ITS)+Bloc+ translation elongation factor-1a (TEF)+RPB1+RPB2). Clades with newly isolated strains are highlighted in orange. Only bootstraps over 80% are shown. A bootstrap of 100% is indicated with an asterisk.

**Figure 3 insects-11-00679-f003:**
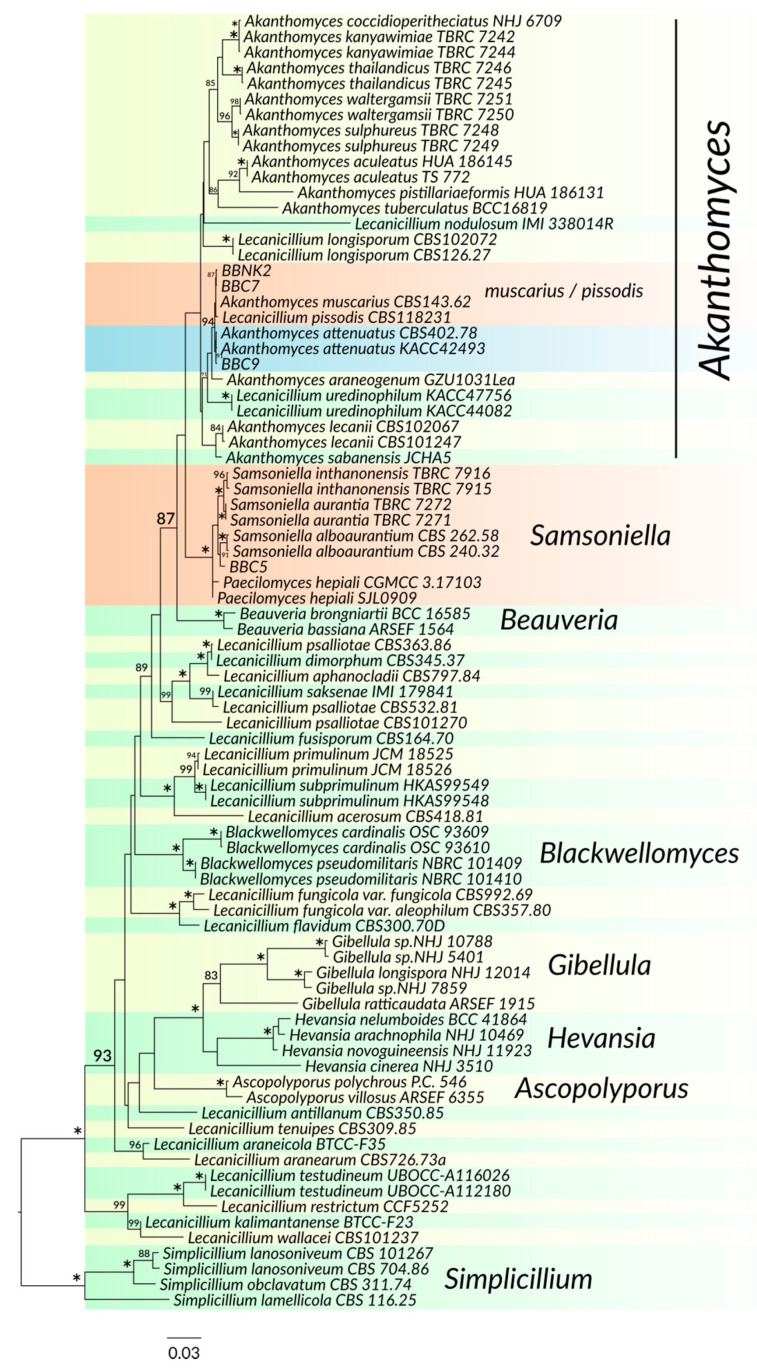
Maximum likelihood phylogenetic tree of the isolated strains BBC5, BBC7, BBC9 and BBCNK2 and related Cordycypitaceae species derived from a combined dataset of ITS+SSU+LSU+TEF+RPB1+RPB2 sequences. Clades with newly isolated strains are highlighted in orange and blue. Only bootstraps over 80% are shown. A bootstrap of 100% is indicated with the asterisk.

**Table 1 insects-11-00679-t001:** Polymerase Chain Reaction (PCR) and sequencing primers used, with references.

Locus	Forward	Reverse	Source
ITS	ITS1F	ITS4	Gardes and Bruns, 1993 [40]; White et al., 1990 [41]
SSU	NS1	NS6	White et al., 1990 [41]
LSU	LR0R	LR5	Vilgalys, unpublished; Vilgalys and Hester, 1990 [42]
TEF	EF1-983F	EF1-2218R	Rehner and Buckley, 2005 [36]
RPB1	CRPB1	RPB1-Cr	Castlebury et al., 2004 [43]; Matheny et al., 2002 [44]
RPB2	RPB2-5F1	RPB2-7cR	Sung et al., 2007 [45]; Reeb et al., 2004 [46]
Bloc	B5.1F	B3.1R	Rehner et al., 2006 [47]

**Table 2 insects-11-00679-t002:** Summary of the main morphological characteristics of four recognized fungal species found on *Corythucha arcuata*.

Isolate	Species	Colonies	Conidiogenous Cell	Conidial Shape and Size
BBC1 BBC2 BBC3 BBC4 BBC6 BBC8 BBC10 BBNK1	*Beauveria pseudobassiana*	white first and becoming yellowish-brown or pale yellow, aerial mycelium dense and farinaceous during sporulation, sometimes producing exudate, slightly changing color of medium to light purple; reverse side yellowish-brown with white margin	conidiophores solitary but usually consisting of dense spherical lateral clusters of globose to flask-shaped conidiogenous cells with an elongating sympodial denticulate rachis giving a zig-zag appearance, 3−6 × 1.3−2.5 μm	globose to subglobose, rarely ellipsoid, 1-celled, hyaline, smooth-walled; 2.5−3.7 × 1.4−3.1 μm
BBC5	*Samsoniella alboaurantium*	white to light orange, cottony and floccose; reverse side deep yellow	conidiophores biverticillate with whorls of flask shaped phialides, 5.7–10 × 1.5–3 μm	conidia lemonshaped, 1-celled, hyaline, smooth walled, 2–3.5 × 1.9–3.5 μm, sometimes in chains
BBC7 BBNK2	*Lecanicillium pissodis*	white; reverse side cream to pale yellow	phialides solitary, or 2 to 3 in whorls, 12.6–38 × 1–2 μm	formed in globose droplets containing up to more than 50 spores, cylindrical to oval, variable in size and shape, 2.6–7.2 × 1.7–2.5 μm
BBC9	*Akanthomyces attenuates*	white; reverse side yellowish-white	up to 3–5 per node, 9.3–15.8 × 1–2 μm	cylindrical with slightly narrowed base, sometimes oval, very variable in size and shape 2.3–6 × 1.5–2.9 μm

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
