# Peer review of "Identification of Entomopathogenic Fungi as Naturally Occurring Enemies of the Invasive Oak Lace Bug, Corythucha arcuata (Say) (Hemiptera: Tingidae)"

_insects, 2020, doi:10.3390/insects11100679_

Round 1
Reviewer 1 Report
Comments and suggestions are given in the ms.
The ms. is the first European report from naturally occurring pathogens of an extremely important invasive alien species Corythucha arcuata. The topic is extremely importan, relevant and actual, since hardyl anything is known about the natural enemies and pathogens of the oak lace bug. The knowledge on natural enemies/pathogens of an invasive species is alway extremely important from pint of assesing its furteher spread/damage and also from point of potential management options.
The topic is original, now similar results have been published from the naturally occurring patheogens of the oak lace bug.
Well written, clearly phased. The English is good enough, but an linguistick check would be useful.
The results are new and important from point of further research related the the oak lace bug. The paper will certainly be frequently cited in the near future.

Author Response
L17: we changed the sentence as requested
L22: we changed the sentence as requested
L29: we changed it as requested
L32: we added the word 'enemies' as requested
L61 : we added the word 'desiccation' (see L65)
L62: we added the word 'yet' (see L67)
L65: we added the word 'and north' (see L71)
L69: We add a sentence and citation of health effects of C. ciliata (see L73-74)
We add the citation in the reference list (see L:385-387)
Dutto, M.; Bertero, M. Dermatosis caused by Corythuca ciliata (Say, 1932) (Heteroptera, Tingidae). diagnostic and clinical aspects of an unrecognized pseudoparasitosis. Journal of Prevenetive Medicine and Hygiene 2009, 54, 57-59.
L71: we deleted a letter 's' (see L78)
L75: we changed 'during the rainy weather' to 'in humid conditions' (see L82)
L93: we changed 'bottom side' to 'underside' (see L115)
L96: we changed 'heavy' to 'severely' (see L118)
L105: infected individuals were collected from several trees, but we have no detailed information to add.
L254: we changed 'habitat' to 'habitats' (see L260)

Reviewer 2 Report
The paper is interesting and data are properly analyzed. Few notes:
why the scientific names are not in italics in the keywords?
P2L46: move the Authority before the taxonomy
L156&245 Cordycypitaceae should be written not in italics
L157 add Authority and classification to Paecilomyces hepiali
Author Response
Why the scientific names are not in italics in the keywords?
-It was a mistake, we corrected it.
P2L46: move the Authority before the taxonomy
-Done (see L50)
L156&245 Cordycypitaceae should be written not in italics
-Done (see L184 and 245)
L157 add Authority and classification to Paecilomyces hepiali
-We add it (see 85-186)
